# How dynamic are ice-stream beds?

Damon Davies[1], Robert G. Bingham[1], Edward C. King[2], Andrew M. Smith[2], Alex M. Brisbourne[2], Matteo Spagnolo[3], Alastair G.C. Graham[4], Anna E. Hogg[5] and David G. Vaughan[2]

[1] School of GeoSciences, University of Edinburgh, Edinburgh, EH8 9XP, UK.
[2] NERC British Antarctic Survey, Madingley Road, Cambridge, CB3 0ET, UK.
[3] School of Geosciences, University of Aberdeen, Aberdeen, AB24 3UF.
[4] College of Life and Environmental Sciences, University of Exeter, Exeter, EX4 4RJ.
[5] Centre for Polar Observation and Modelling, School of Earth and Environment, University of Leeds, Leeds, UK.

*Correspondence to*: Damon Davies (D.Davies@ed.ac.uk)

## Abstract

Projections of sea-level rise contributions from West Antarctica's dynamically thinning ice streams contain high uncertainty because some of the key processes involved are extremely challenging to observe. An especially poorly observed parameter is sub-decadal stability of ice-stream beds, which may be important for subglacial traction, till continuity, and landform development. Only two previous studies have made repeated geophysical measurements of ice-stream beds at the same locations in different years, but both studies were limited in spatial extent. Here, we present the results from repeat radar measurements of the bed of Pine Island Glacier, West Antarctica, conducted 3-6 years apart, along a cumulative ~60 km of profiles. Analysis of the correlation of bed picks between repeat surveys show that 90% of the bed displays no significant change despite the glacier increasing in speed by up to 40% over the last decade. We attribute the negligible detection of morphological change at the bed of Pine Island Glacier to the ubiquitous presence of a deforming till layer, wherein sediment transport is in steady state, such that sediment is transported along the basal interface without inducing morphological change to the radar-sounded basal interface. Given the precision of our measurements, the upper limit of subglacial erosion observed here is 500 mm $a^{-1}$, far exceeding erosion rates reported for glacial settings from proglacial sediment yields, but substantially below subglacial erosion rates of 1.0 m $a^{-1}$ previously reported from repeat geophysical surveys in West Antarctica.

## 1. Introduction

Glaciological studies over the past three decades have revealed that the West Antarctic Ice Sheet (WAIS) is losing mass at an accelerating rate, raising concerns over its potential future contribution to global sea-level (Shepherd et al., 2012, DeConto and Pollard, 2016). Between 2010 and 2013, around 33% of the ice sheet's net mass loss came from Pine Island Glacier (hereafter PIG), a major ice stream draining to the Amundsen Sea Embayment (McMillan et al., 2014). There, from satellite-altimetry observations, mass loss was first expressed at the grounding zone in the mid-1990s, and has now propagated all the way up to the divides (McMillan et al., 2014) by a set of processes broadly termed dynamic thinning (Shepherd et al., 2001; Pritchard et al., 2009). At PIG the dynamic thinning has incorporated an upstream expansion of regions both of accelerating ice (e.g. Scott et al., 2009; Mouginot et al., 2014) and ice-surface lowering (e.g., Konrad et al., 2017), and upstream migration, by several 10s of km, of the grounding line (e.g. Park et al., 2013; Rignot et al., 2014), prompting suggestions that PIG is in runaway retreat (Joughin et al., 2014). However, the detailed processes by which dynamic thinning works, especially upstream from the grounding zone, are not well constrained. Particularly unclear is how the coupling between basal ice and the bed may evolve or vary over time and whether this needs to be accounted for in models projecting future ice response. For example, previous studies have theorised that high subglacial till fluxes can lead to the rapid formation of grounding-zone wedges, potentially stabilising ice streams against sea-level rise (e.g. Alley et al, 1989; Alley et al., 2003; Alley et al., 2007).

Monitoring the beds of ice streams is also important for understanding processes of erosion and sediment transport that can provide information on landscape evolution (e.g., Jamieson et al., 2010, Herman et al., 2011), basal processes (Cuffey and Alley, 1996; Alley et al., 1997; Alley, 2000) and the supply of nutrients to polar oceans (Raiswell et al., 2006). Furthermore, knowledge of till flux and associated till properties is key to an improved understanding of glacier physics and ice-stream stability (Blankenship et al., 1986; Boulton and Hindmarsh, 1987; Alley, 1989; Jenson et al., 1995; Engelhardt and Kamb, 1998; Truffer et al., 2000; Iverson and Iverson, 2001; Nygård et al., 2007; Damsgaard et al., 2013; Damsgaard et al., 2016). A review of previously published erosion rates for hard-bedded glaciers (Hallet et al., 1996) indicates low erosion rates in polar settings (0.01 mm a$^{-1}$) contrasting with relatively high rates beneath temperate alpine glaciers (10-100 mm a$^{-1}$). However, more recent studies have identified far more rapid erosion rates of 4.8 mm a$^{-1}$ in Greenland (Cowton et al., 2012) and as much as 1 m a$^{-1}$ or more in soft-bedded glaciers in Alaska (Motyka et al., 2006) and Antarctica (Smith et al., 2007; 2012). Critically, measurements of erosion rates in Antarctica are limited both temporally and spatially, making it difficult to assess whether such high rates of erosion are typical or exceptional.

A significant difficulty in assessing temporal changes to the beds of WAIS ice streams is lack of access to the bed. Theoretical and geophysical constraints have shown that ice streams typically achieve fast flow by overriding dilated till that provides low basal drag (Alley et al., 1986; Bentley et al., 1998). It has been inferred that even relatively small fluctuations to hydrological conditions in these locations can induce significant changes to basal drag, in the most extreme circumstances causing ice streams to switch on or off (e.g. Anandakrishnan and Alley, 1997; Conway et al., 2002; Vaughan et al., 2008). However, to our knowledge, only two studies have attempted directly to capture temporal changes to ice-stream bed conditions over decadal to sub-decadal timescales commensurate with available satellite records of surface elevation and velocity change. The first was a single-location repeat measurement of ice thickness and surface elevation on PIG (location, Fig. 1) made firstly in 1960 and re-measured in 2009 (Smith et al., 2012). That study found surface lowering with no significant change in ice thickness, suggesting that mean erosion of the ice-stream bed of up to 1 m a$^{-1}$ took place during the study period (Smith et al., 2012). This rate is well above the range reported elsewhere (Hallet et al., 1996). The second, more detailed study, consisted of three repeat-seismic surveys of the bed of Rutford Ice Stream (hereafter RIS; Fig. 1c) obtained in 1991, 1997 and 2004 (Smith et al., 2007). There, across a ~0.5 km width of the bed, 6 m of sediment were removed from the ice bed between 1991 and 1997 followed by the appearance of a drumlin 10 m high and 100 m wide between 1997 and 2004. Both of the studies from West Antarctica therefore implied that active sediment erosion or deposition at rates ~1-1.4 m a$^{-1}$ , and landform evolution, are possible beneath ice streams on decadal to sub-decadal timescales.

In this paper, we present and analyse results from three repeat surveys of the bed of PIG, whereby we geophysically surveyed co-located profiles of PIG's bed in different years along a cumulative ~60 km of traverses. Our principal aim was to ascertain whether erosion or deposition, and any morphological changes, were detectable at the bed of PIG over intervals of 3-7 years, periods over which PIG (from satellite monitoring) has undergone considerable ice-surface lowering and ice acceleration (McMillan et al., 2014, Mouginot et al., 2014).

## 2. Methods

Our data consist of three repeat radar surveys of PIG's bed acquired with a low-frequency ice-penetrating radar system in austral seasons 2007/08 and 2013/14 (two profiles of 18 km and 16 km long separated by ~6 years) and 2010/11 and 2013/14 (one 25 km profile separated by ~3 years) (Figs. 1a,b). The 18 km profile, R1, was acquired 3 km downstream and parallel to an 18 km seismic survey acquired in austral season 2007/08 (Smith et al., 2013), 5 km of which was resurveyed in austral season 2014/15 (Brisbourne et al., 2017) (profiles S1$_{2007}$ and S1$_{2014}$ on Fig. 1b respectively). Each of the repeat-survey locations

on PIG experienced significant ice-surface lowering and acceleration over the encompassing period (Table 1). In our analysis we consider our findings from PIG against the results from Smith et al.'s (2007) repeat-seismic surveys of the bed of RIS (profile C1 in Fig. 1c), where ice flow has remained relatively stable over decadal timescales and negligible surface lowering has been observed (Table 1).

## 2.1 Data acquisition

All radar profiles were acquired with the British Antarctic Survey's "DELORES" (Deep-Look Radio-Echo Sounder) system, a skidoo-towed monopulse array (see King et al., 2016, for general specifications). During the 2007/08 field season, the system operated using 40 m half-dipole antennae, resulting in a centre frequency of 1.2 MHz. In 2010/11 and 2013/14, 20 m half-

dipole antennae were used, giving a centre frequency of 3 MHz. Along-track traces were sampled at <1 m intervals, stacked for noise reduction to produce datapoints at ~5 m spacing, and georeferenced with dual-frequency differential GPS mounted on the radar system.

The repeat-radar surveys on PIG were each acquired approximately orthogonal to ice flow at sites 150, 120 and 115 km

upstream from the 2011 grounding line (Park et al., 2013) (R1, R2 and R3 respectively in Fig. 1b), in each case close to, or forming part of, more extensive radar surveys of patches of the ice bed conducted in the three field seasons (Bingham et al., 2017). Repeat-radar profile R1 comprises a traverse ~18 km in length, first surveyed in January 2008 and then again in December 2013. The January 2008 profile also represents the most downstream traverse of a more extensive set of radar profiles used to image a 108 km$^2$ patch of PIG's bed in January 2008 surrounding the 18 km seismic profile acquired in the

same season (patch 2007t1 and seismic profile S1$_{2007}$ in Fig. 1b). Profile R2 is derived from a 16 km traverse first driven in December 2007 and then again in December 2013; this profile lies ~2 km downstream of an extensive survey of 150 km$^2$ of PIG's bed also surveyed in December 2013 (patch iSTARt1 in Fig. 1b). Profile R3, ~25 km in length, was first surveyed in January 2011 as the upstream profile of multiple transects used to image a 425 km$^2$ patch of PIG's bed (patch 2010tr in Fig. 1b), and then again in December 2013 to yield a survey gap of ~3 years.


**Table 1: Mean velocities for each repeat radar survey profile 2007-2017.**

| Survey | Velocity (m a$^{-1}$) 2007-2009 (MEaSUREs) | Velocity (m a$^{-1}$) 2017* | Velocity change (m a$^{-1}$) | Mean surface elevation change (m)** |
|---|---|---|---|---|
| C1 (RIS) | 370 | 407 | 37 | - |
| R1 (PIG) | 287 | 361 | 74 | -8.2 |
| R2 (PIG) | 384 | 523 | 139 | -8.0 |
| R3 (PIG) | 435 | 609 | 174 | - |

*Velocities derived from Sentinel-1 image pairs obtained in April 2017. Details of processing methodology are provided in Hogg et al., 2017).

**Surface elevation change derived from differential GPS measurements. No GPS data were available for surveys C1 and R3.


## 2.2 Data processing

Radar data were processed using ReflexW seismic processing software (Sandmeier Scientific Software). A data processing flow was applied which included a gain function to improve the strength of reflections at greater depth, and bandpass and 2D median filters to reduce data noise. Finite-difference (FD) migration was used to contract diffraction hyperbolae and to recover

the correct locations of individual reflectors. The onset time of the bed reflector was determined at 5 m horizontal intervals at the peak in the amplitude of the bed reflector using a semi-automated "phase follower" picking procedure that allows automatic

assignment of picks to a selected phase. These picks were checked and edited using manual picking where necessary. Bed picks were then converted to depth using a radar wave speed of 0.168 m ns$^{-1}$ and no additional firn correction We smoothed bed picks by applying a moving average over a 50 m window to all bed picks to remove high-frequency noise.

To assess changes at the bed, we focus on comparing the morphological character of the picked bed along repeat profiles rather than deriving changes in absolute bed elevation between surveys. This is for two reasons. Firstly, we do not have the data to assess whether firn properties, that impact upon radar wave speed, changed over the periods between repeat surveys. However we do not expect firn properties to have varied spatially on the scale of our surveys. For this reason we make no firn correction to our derived ice thicknesses. Secondly, differences in the triggering mechanism of the radar system between survey years
meant that we could not match directly the onset waveforms between repeat surveys. In both cases, the effects preclude recovery of absolute ice thickness or bed elevation. Therefore we compare relative bed profiles by applying a static correction to a common bed datum (0 m) for both surveys.

     The seismic surveys $S1_{2007}$ and $S1_{2014}$ in the vicinity of R1 were processed and analysed by Smith et al. (2013; 2007/08 profile
only) and Brisbourne et al. (2017; both profiles). Brisbourne et al. (2017) primarily report on the calculation of acoustic impedance at the bed from both profiles, which we will consider in our discussion below. For this paper, we also investigated the possibility of directly picking the bed for the repeated 5 km section in an analogous manner to the radar picking described above. However, low signal-noise ratios along large parts of the bed, resulting from the similarity of ice and bed acoustic impedance (Brisbourne et al., 2017), precluded the recovery of results with sufficiently low picking errors to have confidence
in identifying any change or lack thereof.

### 2.3 Errors

For each profile R1-R3, the ability to detect changes in bed morphology is largely determined by the precision with which the bed reflector can be picked, and the degree to which the second radar profile of a repeat survey follows or diverges slightly
from the path driven by the first profile.

     The signal-to-noise ratio in all our radar profiles is high and the strength of the basal reflector produces a clearly discernible, high-amplitude wavelet (e.g. radargrams in Figs. 2-5) that requires little user interaction during the semi-automatic picking procedure. With such clear data, the uncertainty with which the bed reflection can be picked is determined primarily by the
system rise-time of 250 V ns$^{-1}$, the recording-system bandwidth of 50 MHz, and the digitisation interval of 10 ns. Considering also the uncertainty in our GPS-derived elevations, we estimate that our radar data have a vertical range precision of ±3 m.

     As we are unable to recover absolute elevation change, in this study the horizontal resolution is more important than the vertical range precision. Differences in the morphology of the basal reflector need to be considered along with consideration of the
different frequencies used in repeat surveys (1.2 and 3 MHz). This is best illustrated using commonly adopted resolution limits.

For a circular wavefront, features at the bed with a width less than $\sqrt{2d\lambda + \frac{\lambda^2}{4}}$ will appear as point diffractors. For a bed at a depth of 2000 m, a 1.2 MHz ($\lambda_{ice}$=250 m) wavelet will image features with a width <1008 m and a 3 MHz ($\lambda_{ice}$=100 m) will image features with a width <634 m. These differences may affect the appearance of the basal reflector depending on the roughness of the bed. For these reasons we express caution when considering subtle changes in basal morphology.


     Survey lines were repeated by following a route programmed into a dashboard GPS unit mounted on the radar skidoo. Due to the higher accuracy and precision of the dual-frequency GPS compared to the dashboard units, and the challenges of navigating

in featureless terrain, navigational divergences were registered. These divergences were mostly < 50 m (Table 2), however, where bed topography is rough, even small navigational divergences may lead to incorrect interpretation of bed change. In order to visually assess whether navigational divergence affects observed bed change we have provided plots of minimum distance between repeat surveys alongside bed elevation profiles in Figs. 2 and 3.

**Table 2. Analysis of navigational divergence and associated variability in bed elevation.**

| Repeat survey line | R1 | R2 | R3 |
|---|---|---|---|
| **Repeat survey divergence (m)** | | | |
| Maximum | 45.1 | 54.0 | 35.0 |
| Mean | 23.7 | 13.7 | 20.0 |
| Standard Deviation | ±10.3 | ±12.0 | ±8.0 |

## 3. Results

10 Bed picks for the repeat-geophysical surveys from PIG are shown in Figures 2 and 3 alongside associated geophysical images. A visual inspection of the radar/seismic images and bed picks (Figs. 2 and 3) shows that there is remarkable consistency in the morphology of the bed at all three of the repeat-survey sites. This qualitative impression is confirmed by calculating the Pearson correlation coefficient ($r$) of each repeat profile's bed picks across 500 m moving windows (Figs. 2 and 3). For 90% of the repeat radar tracks $r > 0.9$, underscoring that there has been negligible morphological change for much of the surveys. For 15 context, we performed the same correlation routine for the bed picks of repeat-seismic survey data acquired in 1991, 1997 and 2004 along the 3.5 km profile C1 of RIS previously reported by Smith et al. (2007): these results are shown in Fig. 4e. At C1, the ~0.5 km section of track for which Smith et al. (2007) reported basal erosion between 1991 and 1997 yields $r \sim 0.5$, and the ~ 100 m length of profile interpreted as hosting the growth of a drumlin between 1997 and 2004 returns $r \sim -0.2$.

20 There is only one location on PIG where $r$ is considerably lower than 0.9; this occurs between 8.5 and 9 km along profile R1 where $r$ spikes around a value of -0.2 (Fig 2). Closer inspection of this location reveals a subtle change in the morphology of the bed picks between 2007/08 and 2013/14 (Fig. 5). At this site two bumps of ~1-2 m height in 2007/08 are replaced in 2013/14 by a central ridge with two troughs ~3 m in depth. However, it is possible that this change is caused by the aforementioned differences in horizontal resolution of the radar systems between surveys. We are therefore cautious to interpret 25 this as a genuine change in bed morphology resulting from erosion and deposition.

## 4. Discussion

The prevailing picture that emerges from our 58 km of repeat surveys of PIG's bed is one of little measurable change having 30 been effected to the ice stream's basal topography and morphology over the 3-7 year timescale. This is in spite of significant changes occurring to ice-flow speeds and ice-surface elevations over the same periods (Table 1), and the observations of active erosion and deposition made over similar timescales at RIS (Fig. 4), where ice has experienced little to no dynamic thinning and ice flow is essentially stable (Table 1). In the following discussion we firstly consider how these apparently different behaviours between the ice beds of PIG and RIS can be reconciled (Section 4.1). We then turn to the implications of our results

for the understanding of processes of sediment erosion, transport and deposition beneath ice streams and implications for future monitoring of the bed (Section 4.2).

**4.1 Are ice-stream beds dynamic?**

The only precedent for the measurements obtained here from PIG in terms of repeat geophysical survey over comparable timescales is that from RIS where erosion and deposition on the order of ~1 m a$^{-1}$ were observed in association with morphological changes at the bed (Smith et al, 2007). Here we assess whether the lack of comparable changes observed at PIG can be explained by contrasting glaciological, hydrological and basal characteristics to the RIS survey site, and whether either is likely to be more representative of wider changes occurring at the beds of Antarctic ice streams.

The first notable difference between PIG and RIS is the broad subglacial topography. Each of the repeat surveys on PIG was conducted across the 30-km-wide, ~2000 m deep, main ice-stream trunk where, at the multi-km wavelength, the bed is largely flat along and across flow (Vaughan et al., 2006). By contrast, the repeat-survey location on RIS, though also traversing the ice-stream trunk, overlies a notable topographic ridge ("Central low ridge") that abuts ~350 m vertically upwards into the

central ~5 km width of the ~30 km-wide, ~2000 m deep ice-stream trunk (Fig. 4a). There is, therefore, a clear contrast in the gross topographic shape of the cross-sectional bed profile between the PIG and RIS repeat-survey sites; and we note that the flat-bedded trough of PIG is more characteristic of the majority of ice streams in West Antarctica (Fretwell et al., 2013). Turning to the finer, sub-km morphological character of each site, detailed ground-based radar surveys of PIG's trunk (Figs 2 and 3, and see further imagery in Bingham et al., 2017) have depicted ubiquitous mega-scale glacial lineations (MSGL) across

all regions of PIG's trunk, indicative of the widespread presence of deforming sediment (Clark, 1993, Stokes and Clark 2001; Spagnolo et al., 2016). MSGL with mean amplitudes ~10 m and a mode spacing of 300-400 m are also pervasive surrounding the RIS repeat-survey site (Fig. 4a) but, unlike PIG, within the MSGL themselves linear features with much higher amplitudes (up to 70 m) are observed; King et al. (2016) termed these features "tapering drumlins".

The physical properties of the bed between PIG and RIS also differ. Seismic reflection surveying of several sites along PIG's trunk, including upstream and downstream of our repeat-survey sites, has confirmed that the bed immediately below the ice pervasively consists of dilated sediments, which are at least several metres thick (Smith et al., 2013; Brisbourne et al., 2017). Along seismic profile $S1_{2007}$ potential-field data indicate that a transition from sedimentary to crystalline bedrock lies beneath the cap of deformable sediments (at position 9 km on Fig. 2d) (Smith et al., 2013), but the repeat survey of S1 in 2014/15

($S1_{2014}$) exhibited no change to acoustic impedance anywhere along this profile (Brisbourne et al., 2017), reinforcing the notion of a relatively stable basal environment despite the transition in geology below the deforming till layer. By contrast, the bed around RIS profile site C1 is characterised by a "patchwork" of soft, deforming sediments contrasting with regions of basal sliding indicative of more consolidated sediments (Smith and Murray, 2009; Smith et al., 2015). Notably, the areas of deforming sediment coincide with topographic highs such as the feature known as "The Bump" (Smith, 1997) (Fig. 4c).

Changes in acoustic impedance were detected for parts of the RIS repeat-profile between 1997 and 2004, and were interpreted as changes in hydrological conditions within subglacial sediments affecting till porosity (Smith et al., 2007).

A third key difference between the PIG and RIS repeat-survey sites is the degree to which the survey sites are affected by tidal influences, and the potential effects this can have on subglacial hydrology. High-temporal-resolution GPS monitoring of ice

motion at RIS site C1, 10 km upstream of the grounding line, has shown that horizontal ice velocity varies by ~20% on fortnightly timescales in response to tidally-modulated vertical displacement of Filchner-Ronne Ice Shelf (Gudmundsson, 2006, 2007; Murray et al., 2007; Minchew et al., 2017). In addition passive, seismic monitoring has demonstrated increased seismic activity during tidal cycles (Adalgeirsdóttir et al., 2008). Numerical modelling studies have suggested that the velocity

oscillations are transmitted by tidally-forced fluctuations in basal water pressure, that alter the pore pressure of basal sediments and thus effective pressure at the bed (Thompson et al., 2014; Rosier et al., 2015). These observations suggest that the hydrological system beneath the RIS survey site is prone to significant dynamism and reorganisation over short timescales. However, Minchew et al. (2017) suggest that weak shear margins are a more dominant factor in the propagation of tidally-induced horizontal ice-flow variability compared to fluctuations in basal water pressure.

High-resolution GPS monitoring at several sites along PIG's main trunk showed no tidal signal in ice motion even 55 km upstream from the grounding line (Scott et al., 2009), which lies well downstream of our repeat radar sites R1-R3. The ubiquitous dilated till layer that overlies a relatively flat bed at each of the PIG survey sites provides suitable conditions for a stable, distributed drainage system (Weertman, 1972; Alley, 1989; Engelhardt et al., 1990, Engelhardt and Kamb, 1997) potentially in the form of a canal network as suggested in the upstream catchment of neighbouring Thwaites Glacier (Schroeder et al., 2013), upstream RIS (King et al., 2004) and Whillans Ice Stream (Engelhardt and Kamb, 1997). In the absence of a dynamic hydrological system, sediment mobility facilitated by fluvial transport in subglacial sheets or channels (cf. Weertman, 1972; Walder and Fowler, 1994; Fowler 2010, Kyrke-Smith and Fowler, 2014) may be restricted and likely be more stable over time, thereby limiting the rate of erosion and sediment transport detectable within the precision of repeat geophysical measurements.

We therefore consider the possibility that rapid erosion and bed reorganisation on the scale observed beneath RIS is an exception rather than the rule. Surface velocity inversions along PIG's main trunk suggest that most of its bed is subjected to low basal shear stress, except for some discrete "ribs" of high basal traction spanning the trunk downstream from our measurements (Sergienko and Hindmarsh, 2013). We suggest therefore that low sediment transport rates might be expected over much of PIG's bed as a consequence of the generally low basal shear stresses at all of our repeat measurement sites, and that future investigation of bed variation needs to be targeted towards an area of high inferred basal traction.

The apparent stability of the bed we observe is also worth considering in the context of debate concerning strain distribution in deforming beds and associated till rheology. The resolution of our data limit the scope of any firm conclusions but may contribute to further discussion on this issue. Field observations and models have argued for viscous (Boulton and Hindmarsh, 1987; Alley et al., 1987; Hindmarsh, 1998) and plastic (Kamb, 1991; Tulaczyk et al., 2000; Iverson, 2010) deformation of subglacial sediments. Subglacial till transport models invoking either viscous or plastic rheology demonstrate that deformation depth increases with effective pressure. Uneven terrains on an ice stream bed would therefore translate into variable effective pressures and till fluxes, which might facilitate positive feedbacks over bumps and therefore the growth of bedforms (e.g. Hindmarsh, 1998; Fowler, 2000; Schoof, 2007). However, it remains unclear how rapidly bedforms can evolve, as although some studies have suggested rapid growth (Smith et al., 2007; Dowling et al., 2016) these might represent exceptions and a comprehensive analysis is missing. Such pressure-dependent growth will ultimately be controlled by the depth of deformation. It is also unclear whether bedform growth is always limited: they are typically characterised by a relatively well-defined size-frequency distribution, albeit positively skewed (e.g. Fowler et al., 2013; Hillier et al. 2016; Ely et al., 2018). In our study area, one would expect that the uneven terrain of the bed would translate into variable effective pressure and till fluxes and that topography would therefore evolve. The lack of morphological change that we have observed at the ice-stream bed could therefore be interpreted as evidence of very shallow deforming sediment, which might translate to a very low pace of bedform growth, not detectable within the relatively short interval of our repeat surveys. Alternatively, it might indicate that the ice-bed system has reached a point of "maturity" where bedform growth is inhibited by other physical factors. It is even possible that the entire PIG system is now experiencing net erosion due to its recent acceleration, yet the rate of such erosion must be very low for us not to be able to detect a lowering of the topography within the six year interval of our observations.

Geophysical surveys of other West Antarctic ice streams (Alley et al., 1986, Blankenship et al., 1986; Peters et al., 2006) have revealed shallow bed gradients and widespread deforming till similar to the surveyed sites on PIG. These characteristics are also evident in offshore records of palaeo-ice stream beds on the outer continental shelf of West Antarctica, where ice streams occupied shallow troughs in sedimentary basement (Lowe and Anderson, 2002; Wellner et al., 2006; Larter et al., 2009; Graham et al., 2010). These offshore regions are characterised by ubiquitous MSGL (Spagnolo et al., 2014) that are also observed in the more extensive grid surveys surrounding our repeat surveys (Fig. 2a,3a,h) (Bingham et al., 2017). The uniformity of these bedforms may reflect stable, self-organised bed conditions (c.f., Spagnolo et al., 2017).

**4.2 Implications for subglacial sediment transport and future surveys**

The absence of detectable morphological change to the bed over the majority of the ~60 km of bed profiles on PIG could be interpreted in three ways: (1) that no sediment erosion/transport/deposition is occurring at the measured sites; (2) that erosion /deposition is occurring but at rates too low to be detected within the vertical range resolution of the radar; or (3) that the subglacial till flux is in a steady state wherein sediment transport is active but is not altering the shape of the bed. The last of these would contradict modelling studies that suggest that pressure-dependant till flux dictate that there can be no steady-state till flux on an uneven basal interface (Hindmarsh, 1998; Fowler, 2000; Schoof, 2007).

Prior to this study the few repeat geophysical surveys of the ice-bed interface in Alaska and Antarctica yielded subglacial erosion rates of 1000 - 3900 mm $a^{-1}$ (Motyka et al., 2006; Smith et al; 2007; 2012), far exceeding the 0.01 - 100 mm $a^{-1}$ range traditionally reported as characteristic of glacial settings using proglacial sediment yields (Hallet et al., 1996; Koppes and Hallet, 2006; Koppes & Montgomery, 2009; Cowton et al., 2012; Herman et al., 2015) (Fig. 6). Consequently, there has been growing consensus that subglacial erosion and transport is likely to be high, i.e. of the order of m $a^{-1}$, beneath thick, warm-based ice as manifested by polar ice streams. Observations of substantial till deposition at the grounding lines of contemporary and palaeo West Antarctic ice streams support this view (Anandakrishnan et al., 2007; Batchelor and Dowdeswell, 2015). The precision of our measurements, essentially defined by the vertical range resolution of the radar, means that the maximum possible erosion rate that could go undetected along our profiles is 500 mm $a^{-1.}$.

Aside from the repeat geophysical studies conducted in Antarctica by Smith et al. (2007, 2012), the only other location where this method has been used is southeast Alaska where Motyka et al., (2006) found exceptionally high erosion rates of up to 3.9 ± 0.8 m $a^{-1}$ over a period of 14 years (Fig. 6). However, this setting is unique and these exceptional erosion rates occurred during short episodes of glacier advance over glaciomarine and outwash sediments driven by ice-sediment dynamics (Motyka et al., 2006; Brinkerhoff et al., 2017). Indeed, erosion rates in southern Alaska are the highest reported for any region. These high erosion rates are associated with high precipitation rates, active tectonic uplift and glaciofluvial evacuation of unlithified sediments (Hallet et al., 1996; Motyka et al., 2006). The equally rapid erosion observed by Smith et al. (2007, 2012) therefore evokes a glaciofluvial mechanism. Emerging evidence of subglacial hydrology from West Antarctica shows that meltwater can be stored and released over short-timescales through interconnected subglacial lakes and channels (Wingham et al., 2006; Fricker et al., 2007; Fricker and Scambos, 2009; Smith et al., 2017). In our survey area there is no evidence from radar (Bingham et al., 2017) or seismic data (Brisbourne et al., 2017) that indicates the presence of subglacial lakes or concentrated meltwater channels. However, in Smith et al.'s (2012) repeat seismic measurement over ~50 years, rapid erosion of 1 m $a^{-1}$ was detected ~25 km downstream of survey R3 (EHT31 in Fig. 1b). At this location the bed is also smooth and there is no indication from satellite or geophysical data of rapid meltwater drainage events or channelised meltwater flow (Smith et al., 2012). A possible explanation for the discrepancy between erosion observed at this location and the results of this study is a

timescale bias (Ganti et al., 2016). Smith et al.'s (2012) study may have captured intermittent episodes of erosion not captured by our 3-6 year survey intervals.

In Section 3, we showed that while the majority of the repeat-radar profiles evince negligible morphological change, there is one ~0.5 km section of profile R1 where a possible reorganisation of the bed is expressed as a morphological change in the basal reflector (Fig. 5). This region of the bed broadly coincides with a transition from thin ($\leq$ 10 m) sediment overlying a crystalline basement to a deep sedimentary basin imaged in seismic and potential field surveys (Smith et al., 2013). A change in basal drag and ice velocity is also associated with this boundary. It is likely that such boundaries influence subglacial hydrological pathways and, hence, basal ice motion. This exemplifies the importance of the sampling location and periods of measurement to any study attempting to capture subglacial sedimentary processes. For example, the location where Smith et al. (2007) monitored deposition (drumlin formation) on RIS between 1997-2004 is positioned at the end of a drumlin tail subsequently imaged in a wider radar survey of the region (King et al., 2009; 2016) (Fig. 4b). These repeat measurements may have been fortuitously timed to capture the extension of the drumlin tail. The anisotropic morphology of tapering drumlins and MSGL present beneath RIS (King et al., 2016) may not lend itself well to the detection of change from 2-D surveys driven orthogonal to ice flow. Dynamic change in these bedforms may be expressed as downstream migration of a propagating sediment front that would be difficult to detect without more extensive and/or higher-density coverage of repeat surveys over longer timescales.

### 5.0 Conclusions

We have analysed ~60 km of repeat radar surveys acquired from Pine Island Glacier, West Antarctica, between austral seasons 2007/08 and 2013/14. The results showed that little morphological change occurred at the bed over this period. The absence of large signals of erosion or deposition contrasts with erosion/deposition measured on nearby Rutford Ice Stream between 1991 and 2004 (Smith et al., 2007), and with inferred erosion of ~ 1 m a$^{-1}$ at Pine Island Glacier over a half-century (1960-2009; Smith et al., 2012). We attribute the negligible detection of morphological change at the bed of Pine Island Glacier on the sub-decadal timescale to the ubiquitous presence of a deforming till layer, wherein sediment transport is in steady state such that sediment is transported along the basal interface without inducing measurable vertical displacement to the radar-sounded basal interface. Moreover, none of the survey sites on Pine Island Glacier has experienced short-term variations in ice velocity (Scott et al., 2009) diagnostic of active hydrological systems capable of mobilising sediment (Thompson et al., 2014; Rosier et al., 2015). By comparison, the high subglacial erosion and deposition rates reported from Rutford Ice Stream occurred where the ice is overriding a topographic rise 350 m high, there are sharp contrasts in subglacial sediment properties, and the subglacial system likely experiences short-term variability influenced by tidal oscillations (Aðalgeirsdóttir et al., 2008; Thompson et al., 2014; Rosier et al., 2015).

The surveys presented in this study have increased the length, by an order or magnitude, of the available records of repeat measurements of Antarctic ice-stream beds. However, these environments remain poorly sampled. The surveys of Pine Island Glacier and Rutford Ice Stream have shown that improved understanding of the dynamism of the ice-sheet bed may best be gained from a multi-method survey approach, involving: (1) high-density radar grid surveying to image the subglacial landscape and provide spatial context, (2) seismic survey to discern the physical properties of the basal interface, and (3) repeated seismic and/or radar profiles at the same site undertaken over subdecadal to decadal timescales. Our findings also show the importance of the physical properties of the bed at each site to the sediment erosion and deposition that could be detected, underscoring the requirement to sample sites that, as a whole, capture a representative range of basal conditions. Models of basal drag inverted from satellite-imaged surface properties (e.g. Joughin et al., 2009; Arthern et al., 2015) offer the best opportunities to guide site selection.

**Author contributions**

Radar and seismic data from 2007/08 were acquired by RGB; radar data from 2010/11 by ECK; radar data from 2013/14 by DD, RGB, AMS and DGV; and seismic data from 2014/15 by AMB, AMS and DD. Radar data were processed by DD, RGB and ECK and seismic data by AMB and AMS. Sentinel-1 velocity estimates for each geophysical site were produced by AEH. DD analysed the data and wrote the paper. All authors contributed edits to the final manuscript.

**Acknowledgements**

DD was supported by NERC Training Grant NE/K011189/1 to RGB. All fieldwork was supported by funding from the UK Natural Environment Research Council (NERC) iSTAR Programme Grants NE/J005665 (RGB/ECK/AMS/AMB/DGV) and NE/J005681 (AEH), NERC grants NE/B502287 (ECK/AMS) and NE/J004766 (MS), and the British Antarctic Survey (BAS) *Polar Science for Planet Earth* Programme. We would like to thank all members of the iSTAR traverses of Pine Island Glacier for assistance with field-data acquisition in 2013/14 and 2014/15; M. Baird, T. Gee, J. Wake and J. Yates for field-operational support during iSTAR; F. Buckley, C. Griffiths and J. Scott for field assistance in 2007/08 and D. Routledge for field assistance in 2010/11. We are especially grateful for the dedicated and professional support of BAS Operations and Logistics without whom this study would not have been possible. Finally we are grateful to Anders Damsgaard and Huw Horgan for their insightful comments in reviewing this work which has led to a much improved manuscript.