# Peer review of "How dynamic are ice-stream beds?"

_The Cryosphere, 2017_

## Referee Comment (RC1) · A. Damsgaard (Referee) · 14 Nov 2017

Anders Damsgaard

Geophysical Fluid Dynamics Laboratory
Princeton University

**1   General comments**

This paper presents highly valuable results of measurements of subglacial topography evolution, performed by repeat radar measurements on Pine Island Glacier (I am not qualified to evaluate the instrumental procedure and data processing). As outlined in the manuscript, this type of data set only exists for a few locations, and is useful for our understanding of subglacial bed dynamics under changing ice-flow conditions.

The manuscript is well written, and the aim and results are presented in a concise and comprehensible manner. The manuscript clearly deserves publication in *The Cryosphere*. Before publication, I would appreciate if the authors consider the following points:

- Highly relevant to the study area of this manuscript, Sergienko and Hindmarsh (2013) demonstrated large spatial variability in subglacial shear stress inferred from surface-velocity inversion. I would find it very interesting if it was possible to put these results into context of the locations of the proposed "high-shear-stress ribs". I would expect subglacial sediment transport to be low in the areas where stress inversions shows low subglacial friction, regardless of assumed till rheology. Unfortunately it appears that the study area of Sergienko and Hindmarsh (2013) is closer to the grounding line than the site that is investigated here—is that correct? In any case, do your findings present arguments for or against the prior results of localized traction?

- The paper makes several references to changes in ice-surface elevation. If there is any change in bed conditions (bed topography or basal traction), this change should be expressed by a change in surface topography. Opposite, if the bed doesn't change, it should not result in a surface expression. I think it would be very interesting if you could plot surface-elevation profiles at the same points as you have data for the subglacial topography. In case internal reflectors are identifiable and comparable between surveys, a study of these would also bolster the grounds for interpretation.

https://doi.org/10.5194/tc-2017-214

- Strain distribution in deforming beds is a very important and highly controversial topic, tightly connected to the discussion of till rheology. Our understanding of subglacial till fluxes is still lacking, but of primary importance to ice-stream stability, landform development, and glacier physics (Blankenship et al., 1986; Boulton and Hindmarsh, 1987; Alley, 1989; Jenson et al., 1995; Engelhardt and Kamb, 1998; Iverson et al., 1998; Truffer et al., 2000; Iverson and Iverson, 2001; Nygård et al., 2007; Damsgaard et al., 2013; Damsgaard et al., 2016). In till-transport models subglacial sediment flux is dependent on stress and sliding velocity. By coupling sediment transport with a formulation of mass balance (e.g. Exner equation, Exner (1925), Paola and Voller (2005), Fowler (2010), and Kyrke-Smith and Fowler (2014)), ice acceleration by itself will cause a net erosion of the subglacial bed. The nature of subglacial till advection is highly relevant, for example, for theories of drumlin formation (Hindmarsh, 1998; Fowler, 2000; Schoof, 2007; Fowler, 2010). I would find it useful if the results presented in this manuscript are further discussed with relation to implications for till rheology and subglacial advection. I have further elaborated in specific comment 12 below.

I have further comments below, tied to specific page and line numbers. Further referencing, as suggested, may be beneficial to readers less familiar with the topic.

**2   Specific comments**

1. P1L13: How are you doing on the abstract word count relative to journal limits? If you have space, I suggest adding further motivation with something along the lines of: *"An especially poorly observed parameter is sub-decadal stability of ice-stream beds, **which may be important for subglacial traction, till continuity, and landform development.**"*.

2. P1L37: It might be worth mentioning previous studies that hypothesize that large subglacial till fluxes can cause fast build-up of grounding-line wedges, potentially stabilizing ice streams against sea-level rise (e.g. Alley et al., 1989; Alley et al., 2007).

3. P1L40: I suggest also citing Cuffey and Alley (1996) and Alley (2000) which discuss till generation, transport, and associated effects on subglacial beds.

4. P2L1–6: I think it would be helpful to distinguish between soft and hard-bed processes in this paragraph. The cited studies with larger fluxes (e.g. Motyka et al., 2006) discuss reworking of soft beds, while studies with smaller rates (e.g. Hallet et al., 1996) discuss hard-bed erosion. This may not be immediately clear to an outside reader.

5. P2L8–14: Along the lines of my point in "General comments" above, maybe cite and discuss the subglacial stress inversion by Sergienko and Hindmarsh (2013) here.

6. P2L36–39, Table 1, P5L23–24: In several places the text suggests that surface-lowering data is present in Table 1, but the table only includes surface velocities. Is it possible to include the surface-elevation changes in the table?

7. P3L34–P4L2: (I'm a non-expert in glacial radar). Do the same reasons make it impossible to interpret changes in basal reflectivity? As far as I know, changes in the reflectivity have the potential to be an indicator for changes in water content at the ice-bed interface.

8. P5L6–7 and P5L20–21: You previously mentioned that the applied methodology does not make it possible to constrain *absolute* changes in bed elevation, just changes in roughness relative to the mean topography. To me it seems highly unlikely that the bed should have undergone uniform erosion or deposition, but you might want to mention this caveat when presenting the results nonetheless.

9. P6L7–10: Thick porous sediment layers are not necessarily associated with deep active deformation (Damsgaard et al., 2016). See also discrepancy between Blankenship et al. (1986) and Engelhardt et al. (1990).

10. P6L25–27: Minchew et al. (2017) (previously cited in the manuscript) suggests that variations in basal water pressure are likely less important than variations in ice-internal stresses, due to relatively weak rigidity of the RIS ice-stream shear margins. This finding contrasts with the interpretation by Thompson et al. (2014) and Rosier et al. (2015).

11. P6L34–36: It is not immediately clear to me how a dynamic hydrological system would increase sediment transport or erosion. Are you talking about ice-contact subglacial sediment transport or fluvial sediment transport in subglacial sheets or channels (e.g. Alley et al., 2003; Fowler, 2010; Kyrke-Smith and Fowler, 2014)?

12. P7L8–11: Viscous till models and their pressure-dependent fluxes imply that there is no steady-state bed topography if the bed is not planar, as sediment transport would always be larger on the stoss side of bed bumps than on the lee side (Hindmarsh, 1998; Fowler, 2000; Schoof, 2007). Modeling studies indicate similar pressure dependence of till flux for Coulomb-plastic materials (Damsgaard et al., 2013). If there is such a pressure dependence, where increasing normal stress on the bed increases deformational depth, this bed-altering process excludes interpretation (3), *". . . subglacial till flux is in a steady state wherein sediment transport is active but is not altering the shape of the bed."*.

13. P7L23–28: It might be worth mentioning that the site of Alaska is very unusual and that the reported erosion rates by Motyka et al. (2006) are not likely to occur for long. The dynamics of this setting have been further analyzed in Brinkerhoff et al. (2017).

14. P7L40–42, Figure 5: To the untrained eye, it also appears like there has been localized erosion at a distance of 7.6 to 8.0 km. Does this change fall within the bounds of

uncertainty? If not, I think it would make sense to discuss it together with the changes between distances 8.5 to 9.1 km.

15. Figure 1: It took me a moment to realize that the contour lines in panel b are the "dashed lines" referred to in the caption, as the dashes are too small to be apparent on screen and in print. Is it possible to make the dashes and the spacing between them longer?

**References**

Alley, R. B. (1989). "Water-pressure coupling of sliding and bed deformation: II. Velocity-depth profiles". In: *J. Glaciol.* 35.119, pp. 108–118.

— (2000). "Continuity comes first: recent progress in understanding subglacial deformation". In: *Geological Society, London, Special Publications* 176.1, pp. 171–179.

Alley, R. B., D. D. Blankenship, S. T. Rooney, and C. R. Bentley (1989). "Sedimentation beneath ice shelves—the view from ice stream B". In: *Marine Geology* 85.2, pp. 101–120.

Alley, R. B., D. E. Lawson, G. J. Larson, E. B. Evenson, and G. S. Baker (2003). "Stabilizing feedbacks in glacier-bed erosion". In: *Nature* 424.6950, pp. 758–760. doi: `10.1038/nature01839`.

Alley, R. B., S. Anandakrishnan, T. K. Dupont, B. R. Parizek, and D. Pollard (2007). "Effect of sedimentation on ice-sheet grounding-line stability". In: *Science* 315.5820, pp. 1838–1841.

Blankenship, D. D., C. R. Bentley, S. T. Rooney, and R. B. Alley (1986). "Seismic measurements reveal a saturated porous layer beneath an active Antarctic ice stream". In: *Nature*.

Boulton, G. S. and R. C. A. Hindmarsh (1987). "Sediment deformation beneath glaciers: rheology and geological consequences". In: *J. Geophys. Res.* 92.B9, pp. 9059–9082. doi: `10.1029/JB092iB09p09059`.

Brinkerhoff, D., M. Truffer, and A. Aschwanden (2017). "Sediment transport drives tidewater glacier periodicity". In: *Nat. Commun.* 8.1. doi: `10.1038/s41467-017-00095-5`.

Cuffey, K. M. and R. B. Alley (1996). "Is erosion by deforming subglacial sediments significant? (Toward till continuity)". In: *Ann. Glaciol.* 22, pp. 17–24.

Damsgaard, A., D. L. Egholm, J. A. Piotrowski, S. Tulaczyk, N. K. Larsen, and K. Tylmann (2013). "Discrete element modeling of subglacial sediment deformation". In: *J. Geophys. Res. Earth Surf.* 118, pp. 2230–2242. doi: `10.1002/2013JF002830`.

Damsgaard, A., D. L. Egholm, L. H. Beem, S. Tulaczyk, N. K. Larsen, J. A. Piotrowski, and M. R. Siegfried (2016). "Ice flow dynamics forced by water pressure variations in subglacial granular beds". In: *Geophys. Res. Lett.* 43. doi: `10.1002/2016gl071579`.

Engelhardt, H. and B. Kamb (1998). "Basal sliding of ice stream B, West Antarctica". In: *J. Glaciol.* 44.147, pp. 223–230.

Engelhardt, H., N. Humphrey, B. Kamb, and M. Fahnestock (1990). "Physical conditions at the base of a fast moving Antarctic ice stream". In: *Science* 248.4951, pp. 57–59.

Exner, F. M. (1925). "Uber die wechselwirkung zwischen wasser und geschiebe in flüssen". In: *Akad. Wiss. Wien Math. Naturwiss. Klasse* 134.2a, pp. 165–204.

Fowler, A. C. (2000). "An instability mechanism for drumlin formation". In: *Geological Society, London, Special Publications* 176.1, pp. 307–319. doi: `10.1144/GSL.SP.2000.176.01.23`.

— (2010). "The formation of subglacial streams and mega-scale glacial lineations". In: *Proc. R. Soc. Lond. A*. Vol. 466. 2123. The Royal Society, pp. 3181–3201.

Hallet, B., L. Hunter, and J. Bogen (1996). "Rates of erosion and sediment evacuation by glaciers: A review of field data and their implications". In: *Global and Planetary Change* 12.1, pp. 213–235.

Hindmarsh, R. C. A. (1998). "The stability of a viscous till sheet coupled with ice flow, considered at wavelengths less than the ice thickness". In: *J. Glaciol.* 44.147, pp. 285–292. doi: `10.3198/1998JoG44-147-285-292`.

Iverson, N. R. and R. M. Iverson (2001). "Distributed shear of subglacial till due to Coulomb slip". In: *J. Glaciol.* 47.158, pp. 481–488. doi: `10.3189/172756501781832115`.

Iverson, N. R., T. S. Hooyer, and R. W. Baker (1998). "Ring-shear studies of till deformation: Coulomb-plastic behavior and distributed strain in glacier beds". In: *J. Glaciol.* 148, pp. 634–642. doi: `10.3198/1998JoG44-148-634-642`.

Jenson, J. W., P. U. Clark, D. R. MacAyeal, C. Ho, and J. C. Vela (1995). "Numerical modeling of advective transport of saturated deforming sediment beneath the Lake Michigan Lobe, Laurentide Ice Sheet". In: *Geomorphology* 14.2, pp. 157–166.

Kyrke-Smith, T. M. and A. C. Fowler (2014). "Subglacial swamps". In: *Proc. R. Soc. A* 470.2171, p. 20140340. doi: `10.1098/rspa.2014.0340`.

Minchew, B. M., M. Simons, B. Riel, and P. Milillo (2017). "Tidally induced variations in vertical and horizontal motion on Rutford Ice Stream, West Antarctica, inferred from remotely sensed observations". In: *J. Geophys. Res.: Earth Surf.* 122.1, pp. 167–190. doi: `10.1002/2016jf003971`.

Motyka, R. J., M. Truffer, E. M. Kuriger, and A. K. Bucki (2006). "Rapid erosion of soft sediments by tidewater glacier advance: Taku Glacier, Alaska, USA". In: *Geophys. Res. Lett.* 33.24. doi: `10.1029/2006gl028467`.

Nygård, A., H. P. Sejrup, H. Haflidason, W. A. H. Lekens, C. D. Clark, and G. R. Bigg (2007). "Extreme sediment and ice discharge from marine-based ice streams: New evidence from the North Sea". In: *Geology* 35.5, pp. 395–398.

Paola, C. and V. R. Voller (2005). "A generalized Exner equation for sediment mass balance". In: *J. Geophys. Res. Earth Surf.* 110.F4. doi: `10.1029/2004jf000274`.

Rosier, S. H. R., G. H. Gudmundsson, and J. A. M. Green (2015). "Temporal variations in the flow of a large Antarctic ice-stream controlled by tidally induced changes in the subglacial water system". In: *The Cryosphere Discuss.* 9, pp. 2397–2429. doi: `10.5194/tc-9-1649-2015`.

Schoof, C. (2007). "Pressure-dependent viscosity and interfacial instability in coupled ice–sediment flow". In: *J. Fluid Mech.* 570, pp. 227–252. doi: `10.1017/S0022112006002874`.

Sergienko, O. V. and R. C. A. Hindmarsh (2013). "Regular Patterns in Frictional Resistance of Ice-Stream Beds Seen by Surface Data Inversion". In: *Science* 342.6162, pp. 1086–1089.

Thompson, J., M. Simons, and V. C. Tsai (2014). "Modeling the elastic transmission of tidal stresses to great distances inland in channelized ice streams". In: *The Cryosphere* 8.6, pp. 2007–2029. doi: `10.5194/tc-8-2007-2014`.

Truffer, M., W. D. Harrison, and K. A. Echelmeyer (2000). "Glacier motion dominated by processes deep in underlying till". In: *J. Glaciol.* 46.153, pp. 213–221. doi: `10.3189/172756500781832909`.

---

## Referee Comment (RC2) · H. Horgan (Referee) · 22 Nov 2017

**How dynamic are ice-stream beds? Davies et al. November 21st 2017**

Davies et al use repeat radio echo sounding profiles to investigate how the shape of the bed of Pine Island Glacier has changed (or has not changed) over an interval during which the glacier has accelerated and thinned significantly. This study is notable for several reasons. First, Pine Island and the neighboring Thwaites Glacier have experienced substantial mass loss over recent decades and will continue to form a major component of Antarctica's mass balance for the foreseeable future. Second, ground based geophysical observations from this region are rare and provide important constraints on bed structure and properties that influence ice flow. Third, repeat geophysical observation are extremely rare, and are the most practical way of assessing changes at the bed.

This study is topical, well conceived, and generally well presented. The comments I make in the following are intended to support and clarify the manuscript.

**Comparing repeat geophysical surveys.**
Comparing different frequencies of RES is tricky due to the dependence of resolution on frequency. Considering the commonly used resolution limits illustrates the problem. In theory, for a circular wavefront, bed features with a horizontal extent less than $\sqrt{2d\lambda + \frac{\lambda^2}{4}}$ will appear as point diffractors. At these depths (d $\sim$ 2070 m), a 1.2 MHz wavelet will image features smaller than 1025 m as point diffractors, and a 3 MHz wave will do the same for features smaller than 645 m. This is not to say that these features will not be resolved at all, but the bed will appear different at different frequencies depending on how rough it is. Vertical resolution is also frequency dependent. Vertically, layers thinner than $\lambda/30$ will not be resolved (8.3 m at 1.2 MHz, 3.3 m at 3 MHz). This is perhaps not important when considering an abrupt change in dielectric properties at the base of the ice sheet but a layered or gradual bed will resolve differently.

The main conclusion of this paper is, however, that the bed has not changed along the profiles beyond the resolution limits. With a little additional wording regarding the different resolution of the two vintages I have not issue with this conclusion and the resulting interpretation. The reader is, however, drawn to the one location where a change is shown. Resolution differences should be considered here. Also, as pointed out in the manuscript, navigation is a concern. The cross over analysis is useful, but I don't think the method used (mean of the standard deviation of for available intersecting lines) does provide 'the maximum variability in bed elevation'. To add confidence in the interpretation a simple subplot showing minimum distance between the two profiles along the profile would be helpful. This would make it clear to the reader that the region where the bed is different does not correspond to a region of large navigation mismatch.

One last note on the differences between the two data vintages. The 2013/14 data (Fig5c) show more spatial variability in the picks than the 2007/08 data. This may be due simply to signal to noise ratio being lower in the higher frequency 2013/14 data but it would be nice to see a right hand panel showing representative wiggle traces for each of the data vintages. That way the reader could be assured that similar waveforms are being compared.

**Subglacial deformation. Would we expect a change in bed morphology to result from a change in ice surface elevation and ice velocity?**

Estimates of subglacial sediment transport vary widely with the thickness of mobile till and the velocity profile within the till both poorly known. What is accepted is that the till velocity is a function of the overriding ice velocity. Simplistically, a uniform change in ice velocity will result in a uniform change in sediment transport, with no resulting change in bed morphology. If sediment deformation is occurring, as indicated by active source seismic constraints, the total transported through the profile must however have changed and some change in the bed morphology must be taking place upstream and downstream. To address this some discussion of sediment deformation would be useful as would some comment on how uniform the changes in surface elevation and velocity have been.

**Minor Points**

Pg 3 L 28. '5 m intervals' Ambiguous, change to 5 m horizontal intervals

Pg 3 L 29. Please define automation method (e.g. cross correlation).

Pg 3 L 30. 'of 0.168 m ns$^{-1}$' and no additional firn correction.

Pg 3 L 33-37. I can appreciate that the differences in firn composition, and triggering could result in a static shift between the two systems. To clarify this, you should state here that the difference is a constant shift as you don't expect the firn to have varied spatially.

Pg 4 L 1-2. Again, I understand what you have done, but as worded it doesn't as quantitative as it is. You have static corrected both surveys to a common bed-datum, allowing direct comparison.

Pg 4. L 21. '±3 m' Worth a note here that this is not the same as repeatability or resolution. This also depends on what part of the wavelet is being picked. When picking the peak amplitude the wavelet shape will matter. Showing typical wavelets will help with this.

Pg 6. L 22. High temporal resolution

Pg 6. L35. In the absence of a dynamic hydrological system sediment mobility is also likely to be more stable over time.

Pg 7. L 10. 'erosion/deposition' erosion and deposition.

Pg 7. L 19-21. Here's it's probably worth having caveats regarding navigation and the differing resolutions of the two vintages of RES.

Pg 8. L 1. '..'.

Pg 8. L5 'an' a.

Pg 8. L 31. 'seismic survey' seismic surveying.

**Figures**

Figure 1. Is there a good reason to have Antarctica oriented in this direction. We have enough issues with 180 degree flips causing confusion. Caption: 'dashed lines' can't tell they are dashed at this resolution, perhaps gray contours?

Figure 2d should have the same xaxis as b,c,e, and f.

Figure 3. Caption '2km' 2 km

Figure 5a. If it doesn't clutter the figure, can you show the bed prior to smoothing? Figure 5b 5c, show characteristic traces so we can assess waveform similarity and peak amplitude picking suitability.

In closing, I thank the authors for their well considered and presented study.

Sincerely, Huw Horgan

---

## Editor Comment (EC1) · C. R. Stokes (Editor) · 11 Dec 2017

I would like to thank both reviewers for their thoughtful and constructive comments on this manuscript. Given that they are generally very positive, I would certainly encourage the submission of a revised manuscript.

Kind regards,

Chris Stokes
* * *

---

## Author Comment (AC1) · 8 Mar 2018

We thank both reviewers for their insightful and constructive feedback. In the response below we have copied the original comments of reviewers followed by our response in **bold**, original text in grey and changes made to the manuscript in red. Page/line numbers refer to the original manuscript under review at https://doi.org/10.5194/tc-2017-214.

Reviewer 1: Anders Damsgaard

General comments

This paper presents highly valuable results of measurements of subglacial topography evolution, performed by repeat radar measurements on Pine Island Glacier (I am not qualified to evaluate the instrumental procedure and data processing). As outlined in the manuscript, this type of data set only exists for a few locations, and is useful for our understanding of subglacial bed dynamics under changing ice-flow conditions.

The manuscript is well written, and the aim and results are presented in a concise and comprehensible manner. The manuscript clearly deserves publication in *The Cryosphere*.

**We thank the reviewer for his thorough consideration and interest in the manuscript. These comments have helped to improve the manuscript, particularly with regards to implications for till properties and sediment flux, and raising our attention to a number of further interesting manuscripts, many of which we have now referenced in the revised manuscript.**

Before publication, I would appreciate if the authors consider the following points:

Highly relevant to the study area of this manuscript, Sergienko and Hindmarsh (2013) demonstrated large spatial variability in subglacial shear stress inferred from surface velocity inversion. I would find it very interesting if it was possible to put these results into context of the locations of the proposed "high-shear-stress ribs". I would expect subglacial sediment transport to be low in the areas where stress inversions shows low subglacial friction, regardless of assumed till rheology. Unfortunately it appears that the study area of Sergienko and Hindmarsh (2013) is closer to the grounding line than the site that is investigated here— is that correct? In any case, do your findings present arguments for or against the prior results of localized traction?

**Unfortunately our study area does not overlap with these inversions. However we agree that our findings are worth considering in the context of this work.**

**P6 L39 text added as follows:** 'Surface velocity inversions along PIG's main trunk suggest that most of its bed is subjected to low basal shear stress, except for some discrete "ribs" of high basal traction spanning the trunk downstream from our measurements (Sergienko and Hindmarsh, 2013). We suggest therefore that low sediment transport rates might be expected over much of PIG's bed as a consequence of the generally low basal shear stresses at all of our repeat measurement sites, and that future investigation of bed variation needs to be targeted towards an area of high inferred basal traction..'

The paper makes several references to changes in ice-surface elevation. If there is any change in bed conditions (bed topography or basal traction), this change should be expressed by a change in surface topography. Opposite, if the bed doesn't change, it should not result in a surface expression. I think it would be very interesting if you could plot surface-elevation profiles at the same points as you have data for the subglacial topography. In case internal reflectors are identifiable and comparable between surveys, a study of these would also bolster the grounds for interpretation.

> **We have now added plots of surface elevation profiles for surveys R1 and R2. Unfortunately due to problems with the recording of differential GPS surface elevation data for survey R3 in 2010/11 are not available.**

Strain distribution in deforming beds is a very important and highly controversial topic, tightly connected to the discussion of till rheology. Our understanding of subglacial till fluxes is still lacking, but of primary importance to ice-stream stability, landform development, and glacier physics (Blankenship et al., 1986; Boulton and Hindmarsh, 1987; Alley, 1989; Jenson et al., 1995; Engelhardt and Kamb, 1998; Iverson et al., 1998; Truffer et al., 2000; Iverson and Iverson, 2001; Nygård et al., 2007; Damsgaard et al., 2013; Damsgaard et al., 2016). In till-transport models subglacial sediment flux is dependent on stress and sliding velocity. By coupling sediment transport with a formulation of mass balance (e.g. Exner equation, Exner (1925), Paola and Voller (2005), Fowler (2010), and Kyrke-Smith and Fowler (2014)), ice acceleration by itself will cause a net erosion of the subglacial bed. The nature of subglacial till advection is highly relevant, for example, for theories of drumlin formation (Hindmarsh, 1998; Fowler, 2000; Schoof, 2007; Fowler, 2010). I would find it useful if the results presented in this manuscript are further discussed with relation to implications for till rheology and subglacial advection. I have further elaborated in specific comment 12 below.

> **These are all excellent points and we have added the following to the introduction and discussion.**
>
> **P2 L1-4 text added as follows:** 'Furthermore, knowledge of till flux and associated till properties is key to an improved understanding of glacier physics and ice-stream stability (Blankenship et al., 1986; Boulton and Hindmarsh, 1987; Alley, 1989; Jenson et al., 1995; Engelhardt and Kamb, 1998; Truffer et al., 2000; Iverson and Iverson, 2001; Nygård et al., 2007; Damsgaard et al., 2013; Damsgaard et al., 2016).'
>
> **P7 L28-46 text added as follows:** 'The apparent stability of the bed we observe is also worth considering in the context of debate concerning strain distribution in deforming beds and associated till rheology. The resolution of our data limit the scope of any firm conclusions but may contribute to further discussion on this issue. Field observations and models have argued for viscous (Boulton and Hindmarsh, 1987; Alley et al., 1987; Hindmarsh, 1998) and plastic (Kamb, 1991; Tulaczyk et al., 2000; Iverson, 2010) deformation of subglacial sediments. Subglacial till transport models invoking either viscous or plastic rheology demonstrate that deformation depth increases with effective pressure. Uneven terrains on an ice stream bed would therefore translate into variable effective pressures and till fluxes, which might facilitate positive feedbacks over bumps and therefore the growth of bedforms (e.g. Hindmarsh, 1998; Fowler, 2000; Schoof, 2007). However, it remains unclear how rapidly bedforms can evolve, as although some studies have suggested rapid growth (Smith et al., 2007; Dowling et al., 2016) these might represent exceptions and a comprehensive analysis is missing. Such pressure-dependent growth will ultimately be controlled by the depth of deformation. It is also unclear whether bedform growth is always limited: they are typically characterised by a relatively well-defined size-frequency distribution, albeit positively skewed (e.g. Fowler et al., 2013; Hillier et al. 2016; Ely et al., 2018;). In our study area, one would expect that the uneven terrain of the bed would translate into variable effective pressure and till fluxes and that topography would therefore evolve. The lack of morphological change that we have observed at the ice-stream bed could therefore be interpreted as evidence of very shallow deforming sediment, which might translate to a very low pace of bedform growth, not detectable within the relatively short interval of our repeat surveys. Alternatively, it might indicate that the ice-bed system has reached a point of "maturity" where bedform growth is inhibited by other physical factors.

It is even possible that the entire PIG system is now experiencing net erosion due to its recent acceleration, yet the rate of such erosion must be very low for us not to be able to detect a lowering of the topography within the six year interval of our observations..'

I have further comments below, tied to specific page and line numbers. Further referencing, as suggested, may be beneficial to readers less familiar with the topic.

Specific comments

1. P1L13: How are you doing on the abstract word count relative to journal limits? If you have space, I suggest adding further motivation with something along the lines of: *"An especially poorly observed parameter is sub-decadal stability of ice-stream beds, **which may be important for subglacial traction, till continuity, and landform development.**"*.

**We may be pushing the word limit but have included the text.**

**P1 L13-14 modified as follows** 'An especially poorly observed parameter is sub-decadal stability of ice-stream beds, which may be important for subglacial traction, till continuity, and landform development."

2. P1L37: It might be worth mentioning previous studies that hypothesize that large subglacial till fluxes can cause fast build-up of grounding-line wedges, potentially stabilizing ice streams against sea-level rise (e.g. Alley et al., 1989; Alley et al., 2007).

**P1 L38-40 text added as follows:** 'For example, previous studies have theorised that high subglacial till fluxes can lead to the rapid formation of grounding-zone wedges, potentially stabilising ice streams against sea-level rise (e.g. Alley et al, 1989; Alley et al., 2003; Alley et al., 2007).'

3. P1L40: I suggest also citing Cuffey and Alley (1996) and Alley (2000) which discuss till generation, transport, and associated effects on subglacial beds.

**P2 L3 modified as follows:** Monitoring the beds of ice streams is also important for understanding processes of erosion and sediment transport that can provide information on landscape evolution (e.g., Jamieson et al., 2010, Herman et al., 2011), basal processes (Cuffey and Alley, 1996; Alley et al., 1997; Alley, 2000)…'

4. P2L1–6: I think it would be helpful to distinguish between soft and hard-bed processes in this paragraph. The cited studies with larger fluxes (e.g. Motyka et al., 2006) discuss reworking of soft beds, while studies with smaller rates (e.g. Hallet et al., 1996) discuss hard-bed erosion. This may not be immediately clear to an outside reader.

**P2 L8-11 modified as follows:** 'A review of previously published erosion rates for hard-bedded glaciers (Hallet et al., 1996) indicates low erosion rates in polar settings (0.01 mm a-1) contrasting with relatively high rates beneath temperate alpine glaciers (10-100 mm a-1). However, more recent studies have identified far more rapid erosion rates of 4.8 mm a-1 in Greenland (Cowton et al., 2012) and as much as 1000 mm a-1 or more in soft-bedded glaciers in Alaska (Motyka et al., 2006) and Antarctica (Smith et al., 2007; 2012).'

5. P2L8–14: Along the lines of my point in "General comments" above, maybe cite and discuss the subglacial stress inversion by Sergienko and Hindmarsh (2013) here.

**See response to general comments**

6. P2L36–39, Table 1, P5L23–24: In several places the text suggests that surface-lowering data is present in Table 1, but the table only includes surface velocities. Is it possible to include the surface-elevation changes in the table?

> **We have added mean surface elevation change data for surveys R1 and R2. For the same reason as we provided in response to the general comment regarding surface elevation profiles, data are not available for survey R3.**

7. P3L34–P4L2: (I'm a non-expert in glacial radar). Do the same reasons make it impossible to interpret changes in basal reflectivity? As far as I know, changes in the reflectivity have the potential to be an indicator for changes in water content at the ice-bed interface.

> **In addition to the differences in the radar system configuration between years outlined in the methods section, we have found that reflectivity (or bed returned power) can be affected by the system battery power. For these reasons we are not confident over the intervals concerned of interpreting any changes in bed returned power to represent changes in water content at the basal interface.**

8. P5L6–7 and P5L20–21: You previously mentioned that the applied methodology does not make it possible to constrain *absolute* changes in bed elevation, just changes in roughness relative to the mean topography. To me it seems highly unlikely that the bed should have undergone uniform erosion or deposition, but you might want to mention this caveat when presenting the results nonetheless.

> **We are of the view that the possibility of uniform erosion and deposition along a cumulative 60 km of survey line is so unlikely that it renders the inclusion of such a caveat unnecessary.**

9. P6L7–10: Thick porous sediment layers are not necessarily associated with deep active deformation (Damsgaard et al., 2016). See also discrepancy between Blankenship et al. (1986) and Engelhardt et al. (1990).

> **We agree that this distinction is important with regards to sediment properties. We have changed the wording to avoid the inference that the thick, dilated sediments are necessarily deforming.**
>
> **P6 L16-19 modified as follows:** '…immediately below the ice pervasively consists of dilated sediments, which are at least several metres thick (Smith et al., 2013; Brisbourne et al., 2017). Along seismic profile S12007 potential-field data indicate that a transition from sedimentary to crystalline bedrock lies beneath the cap of deformable sediments…'

10. P6L25–27: Minchew et al. (2017) (previously cited in the manuscript) suggests that variations in basal water pressure are likely less important than variations in ice internal stresses, due to relatively weak rigidity of the RIS ice-stream shear margins.
This finding contrasts with the interpretation by Thompson et al. (2014) and Rosier et al. (2015).

> **P6 L36-38 modified as follows:** 'However, Minchew et al. (2017) suggest that weak shear margins are a more dominant factor in the propagation of tidally-induced horizontal ice-flow variability compared to fluctuations in basal water pressure.'

11. P6L34–36: It is not immediately clear to me how a dynamic hydrological system would increase sediment transport or erosion. Are you talking about ice-contact subglacial sediment transport or fluvial sediment transport in subglacial sheets or channels (e.g. Alley et al., 2003; Fowler, 2010; Kyrke-Smith and Fowler, 2014)?

**We were referring to fluvial sediment transport but we agree this is not clear from the text.**

**P7 L2-5 modified as follows: '**In the absence of a dynamic hydrological system, sediment mobility facilitated by fluvial transport in subglacial sheets or channels (cf. Weertman, 1972; Walder and Fowler, 1994; Fowler 2010, Kyrke-Smith and Fowler, 2014) may be restricted, thereby limiting the rate of erosion and sediment transport detectable within the precision of repeat geophysical measurements.

12. P7L8–11: Viscous till models and their pressure-dependent fluxes imply that there is no steady-state bed topography if the bed is not planar, as sediment transport would always be larger on the stoss side of bed bumps than on the lee side (Hindmarsh, 1998; Fowler, 2000; Schoof, 2007). Modeling studies indicate similar pressure dependence of till flux for Coulomb-plastic materials (Damsgaard et al., 2013). If there is such a pressure dependence, where increasing normal stress on the bed increases deformational depth, this bed-altering process excludes interpretation (3), *"…subglacial till flux is in a steady state wherein sediment transport is active but is not altering the shape of the bed."*.

**Because all our repeat surveys are oriented transverse to flow we are unable to assess stoss/lee variability in bedform morphology. We do not discount pressure dependent sediment flux. We are referring to the cross-sectional morphology only. We have included additional text to acknowledge**

**P7 L28-30 text added as follows: '**The last of these would contradict modelling studies with pressure-dependent till flux that dictate that there can be no steady-state till flux on an uneven basal interface (Hindmarsh, 1998; Fowler, 2000; Schoof, 2007).'

13. P7L23–28: It might be worth mentioning that the site of Alaska is very unusual and that the reported erosion rates by Motyka et al. (2006) are not likely to occur for long. The dynamics of this setting have been further analyzed in Brinkerhoff et al. (2017).

**P8 L1-4 modified as follows: '**However, this setting is unique and these exceptional erosion rates occurred during short episodes of glacier advance over glaciomarine and outwash sediments driven by ice-sediment dynamics (Motyka et al., 2006; Brinkerhoff et al., 2017). Indeed, erosion rates in southern Alaska are the highest reported for any region.'

14. P7L40–42, Figure 5: To the untrained eye, it also appears like there has been localized erosion at a distance of 7.6 to 8.0 km. Does this change fall within the bounds of uncertainty? If not, I think it would make sense to discuss it together with the changes between distances 8.5 to 9.1 km.

**We are not confident that the changes we discuss in the results are genuine alterations of bed morphology. Additional caveats to our ability to detect changes at the bed given the differences in the radar systems have been included in response to comments from reviewer 2. We have also include additional text in the results section:**

**P5 L18-20 modified as follows: '**However, it is possible that this change is caused by the aforementioned differences in frequency of the radar systems between surveys. We are

therefore cautious to interpret this as a genuine change in bed morphology resulting from erosion and deposition.'

15. Figure 1: It took me a moment to realize that the contour lines in panel b are the "dashed lines" referred to in the caption, as the dashes are too small to be apparent on screen and in print. Is it possible to make the dashes and the spacing between them longer?

**We have changed the figure caption to describe velocity contours as "grey lines".**

---

## Author Comment (AC2) · 8 Mar 2018

We thank both reviewers for their insightful and constructive feedback. In the response below we have copied the original comments of reviewers followed by our response in **bold**, original text in grey and changes made to the manuscript in red. Page/line numbers refer to the original manuscript under review at https://doi.org/10.5194/tc-2017-214.

**Reviewer 2: Huw J. Horgan**

Davies et al use repeat radio echo sounding profiles to investigate how the shape of the bed of Pine Island Glacier has changed (or has not changed) over an interval during which the glacier has accelerated and thinned significantly. This study is notable for several reasons. First, Pine Island and the neighboring Thwaites Glacier have experienced substantial mass loss over recent decades and will continue to form a major component of Antarctica's mass balance for the foreseeable future. Second, ground based geophysical observations from this region are rare and provide important constraints on bed structure and properties that influence ice flow. Third, repeat geophysical observation are extremely rare, and are the most practical way of assessing changes at the bed.

This study is topical, well conceived, and generally well presented. The comments I make in the following are intended to support and clarify the manuscript.

> **We are grateful to the reviewer for his comments and for enabling us to improve the clarity of the technical aspects relating to the radar systems.**

Comparing repeat geophysical surveys. Comparing different frequencies of RES is tricky due to the dependence of resolution on frequency. Considering the commonly used resolution limits illustrates the problem. In theory, for a circular wavefront, bed features with a horizontal extent less than $\sqrt{2d\lambda + \frac{\lambda^2}{4}}$ will appear as point diffractors. At these depths (d ~ 2070 m), a 1.2 MHz wavelet will image features smaller than 1025 m as point diffractors, and a 3 MHz wave will do the same for features smaller than 645 m. This is not to say that these features will not be resolved at all, but the bed will appear different at different frequencies depending on how rough it is. Vertical resolution is also frequency dependent. Vertically, layers thinner than λ/30 will not be resolved (8.3 m at 1.2 MHz, 3.3 m at 3 MHz). This is perhaps not important when considering an abrupt change in dielectric properties at the base of the ice sheet but a layered or gradual bed will resolve differently.

The main conclusion of this paper is, however, that the bed has not changed along the profiles beyond the resolution limits. With a little additional wording regarding the different resolution of the two vintages I have not issue with this conclusion and the resulting interpretation. The reader is, however, drawn to the one location where a change is shown. Resolution differences should be considered here. Also, as pointed out in the manuscript, navigation is a concern. The cross over analysis is useful, but I don't think the method used (mean of the standard deviation of for available intersecting lines) does provide `the maximum variability in bed elevation'. To add confidence in the interpretation a simple subplot showing minimum distance between the two profiles along the profile would be helpful. This would make it clear to the reader that the region where the bed is different does not correspond to a region of large navigation mismatch.

> **Based on the above, we have expanded on the caveats with regards to the differences in the frequency of the radar systems in the section 2.3.**
>
> **P4 L31-37 modified as follows:** 'As we are unable to recover absolute elevation change, in this study the horizontal resolution is more important than the vertical range precision. Differences in the morphology of the basal reflector need to be considered along with

consideration of the different frequencies used in repeat surveys (1.2 and 3 MHz). This is best illustrated using commonly adopted resolution limits. For a circular wavefront, features at the bed with a width less than $\sqrt{2d\lambda + \frac{\lambda^2}{4}}$ will appear as point diffractors. For a bed at a depth of 2000 m, a 1.2 MHz ($\lambda_{ice}$=250 m) wavelet will image features with a width <1008 m and a 3 MHz ($\lambda_{ice}$=100 m) will image features with a width <634 m. These differences may affect the appearance of the basal reflector depending on the roughness of the bed. For these reasons we express caution when considering subtle changes in basal morphology.'

**With regards to comments concerning navigational divergence, we have removed the crossover analysis from the table and text and added profiles showing the minimum distance between survey profiles alongside plots of bed elevation and bed pick correlation in Figs. 2 and 3.**

**P5 L1-5 removed the following text:** "We assessed the variability in bed topography by analysing nine additional intersecting radar lines driven orthogonal to repeat survey lines (three per repeat survey line). We calculated a mean of the standard deviation of bed elevation for available intersecting lines within 50 m, east and west of repeat survey lines (Table 2). From this analysis we find that the maximum variability in bed elevation within the range of navigational divergence between repeat surveys is just over a metre (Table 2)."

**P5 L1 added the following text: "**In order to visually assess whether navigational divergence affects observed bed change we have provided plots of minimum distance between repeat surveys alongside plots alongside bed elevation profiles in Figs. 2 and 3."

One last note on the differences between the two data vintages. The 2013/14 data (Fig5c) show more spatial variability in the picks than the 2007/08 data. This may be due simply to signal to noise ratio being lower in the higher frequency 2013/14 data but it would be nice to see a right hand panel showing representative wiggle traces for each of the data vintages. That way the reader could be assured that similar waveforms are being compared.

**We have added inset plots of representative wiggle traces of 2007/08 and 2013/14 data to figure 5b and c.**

**Subglacial deformation. Would we expect a change in bed morphology to result from a change in ice surface elevation and ice velocity?**

Estimates of subglacial sediment transport vary widely with the thickness of mobile till and the velocity profile within the till both poorly known. What is accepted is that the till velocity is a function of the overriding ice velocity. Simplistically, a uniform change in ice velocity will result in a uniform change in sediment transport, with no resulting change in bed morphology. If sediment deformation is occurring, as indicated by active source seismic constraints, the total transported through the profile must however have changed and some change in the bed morphology must be taking place upstream and downstream. To address this some discussion of sediment deformation would be useful as would some comment on how uniform the changes in surface elevation and velocity have been.

We have responded on similar points that were raised by Reviewer #1 – see our responses to the last "General Comments" from Reviewer #1 above.

**Minor Points**
Pg 3 L 28. `5 m intervals' Ambiguous, change to 5 m horizontal intervals

This change is incorporated into the next point.

Pg 3 L 29. Please define automation method (e.g. cross correlation).

**P3 L35–37 modified as follows:** 'The onset time of the bed reflector was determined at 5 m horizontal intervals at the peak in the amplitude of the bed reflector using a semi-automated "phase follower" picking procedure that allows automatic assignment of picks to a selected phase. These picks were checked and edited using manual picking where necessary.'

Pg 3 L 30. `of 0.168 m ns$^{-1}$' and no additional firn correction.

Done

Pg 3 L 33-37. I can appreciate that the differences in firn composition, and triggering could result in a static shift between the two systems. To clarify this, you should state here that the difference is a constant shift as you don't expect the firn to have varied spatially.

**P4 L5-7 modified as follows:** 'Firstly, we do not have the data to assess whether firn properties, that impact upon radar wave speed, changed over the periods between repeat surveys. However we do not expect firn properties to have varied spatially on the scale of our surveys.'

Pg 4 L 1-2. Again, I understand what you have done, but as worded it doesn't as quantitative as it is. You have static corrected both surveys to a common bed-datum, allowing direct comparison.

**P4 L10-11 modified as follows:** 'Therefore we compare relative bed profiles by applying a static correction to a common bed datum (0 m) for both surveys.'

Pg 4. L 21. `±3 m' Worth a note here that this is not the same as repeatability or resolution. This also depends on what part of the wavelet is being picked. When picking the peak amplitude the wavelet shape will matter. Showing typical wavelets will help with this.

**We have addressed this comment by adding subplots of wiggle traces to address the previous general comment.**

Pg 6. L 22. High temporal resolution

**Done**

Pg 6. L 35. In the absence of a dynamic hydrological system sediment mobility is also likely to be more stable over time.

**P7 L16-17 modified as follows:** 'may be restricted and likely be more stable over time, thereby limiting…'

Pg 7. L 10. `erosion/deposition' erosion and deposition.

**Done.**

Pg 7. L 19-21. Here's it's probably worth having caveats regarding navigation and the differing resolutions of the two vintages of RES.

> **We have addressed this comment by adding more text to the methods section to clarify challenges with comparing different frequencies and added wiggle trace plots to Fig. 5.**

Pg 8. L 1. `..'.
Pg 8. L5 `an' a.
Pg 8. L 31. 'seismic survey' seismic surveying.

> **All done.**

**Figures**

Figure 1. Is there a good reason to have Antarctica oriented in this direction. We have enough issues with 180 degree ips causing confusion.

> **Our figures are oriented to have true north pointing vertically upwards over PIG, following a convention typical in many maps. This orientation is also used in several other papers concerned with the Amundsen Sea Embayment.**

Caption: `dashed lines' can't tell they are dashed at this resolution, perhaps gray contours?

> **Figure caption modified as suggested.**

Figure 2d should have the same x-axis as b,c,e, and f.

> **Figure axis modified to the same labelling format.**

Figure 3. Caption `2km' 2 km

> **Changed.**

Figure 5a. If it doesn't clutter the figure, can you show the bed prior to smoothing?

> **We feel this will clutter to the figure so would prefer to leave this as shown.**

Figure 5b 5c, show characteristic traces so we can assess waveform similarity and peak amplitude picking suitability.

> **Done.**

In closing, I thank the authors for their well considered and presented study.

Sincerely, Huw Horgan